# Biological Evaluation of Azetidine-2-One Derivatives of Ferulic Acid as Promising Anti-Inflammatory Agents

**Maria Drăgan [1], Cătălina Daniela Stan [1,\*], Andreea Teodora Iacob [2] , Oana Maria Dragostin [3], Mihaela Boancă [4] , Cătălina Elena Lupuşoru [4], Carmen Lăcrămioara Zamfir [5] and Lenuţa Profire [2]**

[1] Department of Drugs Industry and Pharmaceutical Biotechnology, Faculty of Pharmacy, Grigore T. Popa University of Medicine and Pharmacy, 16 Universitatii Str., 700115 Iasi, Romania; maria.wolszleger@umfiasi.ro

[2] Department of Pharmaceutical Chemistry, Faculty of Pharmacy, Grigore T. Popa University of Medicine and Pharmacy, 16 Universitatii Str., 700115 Iasi, Romania; panzariu.andreea.teodora@gmail.com (A.T.I.); lenuta.profire@umfiasi.ro (L.P.)

[3] Faculty of Medicine and Pharmacy, Dunarea de Jos University of Galati, 47 Domneasca Str., 800008 Galati, Romania; Oana.Dragostin@ugal.ro

[4] Department of Pharmacology, Faculty of Medicine, Grigore T. Popa University of Medicine and Pharmacy, 16 Universitatii Str., 700115 Iasi, Romania; boanca.mihaela@umfiasi.ro (M.B.); lupusorucatalina@gmail.com (C.E.L.)

[5] Department of Histology, Faculty of Medicine, Grigore T. Popa University of Medicine and Pharmacy, 16 Universitatii Str., 700115 Iasi, Romania; carmen.zamfir@umfiasi.ro

\* Correspondence: catalina.stan@umfiasi.ro

**Abstract:** The purpose of this study was to evaluate the in vivo biological potential of new azetidine-2-one derivatives of ferulic acid (**6a–f**). First, the in vivo acute toxicity of azetidine-2-one derivatives of ferulic acid on Swiss white mice was investigated and, based on the obtained results, it can be stated that the studied derivatives belong to compounds with moderate toxicity. The in vivo anti-inflammatory potential of these derivatives was determined in a model of acute inflammation induced by carrageenan in rats and in a chronic inflammation model induced in rats using the granuloma test. In the acute inflammation model, all the studied compounds had a maximum anti-inflammatory effect 24 h after administration, which suggests that these compounds may be classified, from a pharmacokinetic point of view, in the category of long-acting compounds. The most active compound in the series was found to be compound **6b**. In the case of the chronic inflammation model, it was observed that the studied compounds (**6a–f**) reduced the formation of granulation tissue compared to the control group, having an intense effect of inhibiting the proliferative component. The most important inhibitory effect of inhibiting the proliferative component was recorded for compound **6b**. Additionally, the investigation of liver function was performed by determining the serum levels of liver enzymes aspartate transaminase (AST), alanine aminotransferase (ALT), lactate dehydrogenase (LDH) and bilirubin (total and direct). The results showed that, in the series of azetidin-2-one derivatives, the liver enzymes concentration values were close to those recorded for the reference anti-inflammatories (diclofenac sodium and indomethacin) and slightly higher compared to the values for the healthy control group. At the end of the experiment, the animals were euthanized and fragments of liver, lung, and kidney tissue were taken from all groups in the study. These were processed for histopathological examination, and we noticed no major changes in the groups treated with the azetidine 2-one derivatives of ferulic acid compared to the healthy groups.

**Keywords:** azetidine-2-one derivatives; ferulic acid; acute inflammation; chronic inflammation; biochemical parameters; histopathological study

## 1. Introduction

Free radical damage leads to oxidative stress-related disorders, including inflammatory diseases [1].

An inflammatory process in cellular and tissue levels is an important stress factor for the human organism which can trigger chronic diseases, including diabetes, cancer, cardiovascular diseases, arthritis, obesity, and autoimmune diseases.

Inflammation is a complex defense mechanism to harmful stimuli from different biological and environmental sources in order to restore an injured tissue by the inflammation cascade [2].

Depending on the involved immune factors and the duration of the process, inflammation has been classified as an acute and chronic process. Acute inflammation is caused by cellular and vascular reactions which are responsible for the clinical symptoms of inflammation, and chronic inflammation occurs when the human organism is exposed to foreign bodies, chemical agents, and specific pathogens [2].

Therefore, antioxidant and anti-inflammatory agents may be useful as valuable drugs for inflammatory diseases [1].

Ferulic acid is a ubiquitous plant constituent that is one of the most attractive secondary metabolites. Is a phenolic compound found in the cell wall of plant tissues, where it is conjugated with mono-, di-, and polysaccharides. It has proved to be a potent antioxidant and a very good free-radical scavenger due to its electron-donating substituents, thus ameliorating oxidative stress.

Additionally, it was found that ferulic acid has beneficial effects against cancer, diabetes, neurodegenerative diseases, and the cell aging process and is able to protect human skin from UV irradiation [3].

On the other hand, heterocycles are one of the most significant classes of compounds which have been investigated in pharmaceutical research areas. The azetidin-2-one ring (as an important heterocycle) has drawn the attention of scientists and has been evaluated and investigated over the years. To start with, azetidin-2-ones were explored for antibacterial activity. As a result of those studies and being endowed with a unique structure, azetidin-2-ones proved to be potent antibacterials (β-lactam antibiotics), including penicillins, cephalosporins, carbapenems, monobactams, clavulanic acid, sulbactam, and tazobactam [4]. Recently, the azetidin-2-one scaffold has provided compounds with a wide spectrum of activities, such as cholesterol absorption inhibitory (Ezetimibe), anticancer, antitubercular, antidiabetic, anti-inflammatory, analgesic, thrombin inhibition, antiparkinsonian, vasopressin antagonist, anticonvulsant, enzyme inhibitors [5,6].

Starting from these data, we previously obtained by cyclisation six new azetidine-2-one derivatives of ferulic acid which are shown in Table 1.

**Table 1.** Azetidin-2-one derivatives of ferulic acid (**6a–f**).

**6(a–f)**

| Compound | –R | Compound | –R |
|----------|------|----------|------------|
| **6a** | –H | **6d** | –NO$_2$(2) |
| **6b** | –F(4) | **6e** | –Br(4) |
| **6c** | –Cl(4) | **6f** | –OH(2) |

The synthesis, spectral characterization, in vitro antioxidant and anti-inflammatory potential, and antimicrobial action have already been published [7,8].

In those studies, the in vitro antioxidant potential of the azetidin-2-one derivatives of ferulic acid was assessed using the total antioxidant capacity, total reducing power assays, and DPPH and ABTS$^{.+}$ radicals scavenging assays. The results showed that all the investigated compounds possess a good antioxidant activity at low concentrations, a more intense effect than ferulic acid and a comparable effect with ascorbic acid, used as positive control, was found for compound **6b** (R = 4-F) [7,8]. Additionally, the in vitro anti-inflammatory potential was investigated using bovine serum albumin denaturation and human red blood cell membrane stabilization assays. The same compound **6b** (R = 4-F) proved to be the most active, with the anti-denaturation activity being more intense than that of ferulic acid and comparable with the diclofenac used as a positive control at the concentration of 50 µg/mL. Additionally, this compound showed a membrane stabilizing activity higher than ferulic acid and comparable with diclofenac, especially at a 1100 µg/mL concentration [8].

Therefore, the aim of this study was to evaluate the in vivo anti-inflammatory potential of the azetidin-2-one derivatives of ferulic acid, taking into account their acute toxicity. Additionally, we evaluate the liver function by determining the serum levels of liver enzymes (AST, ALT, LDH) and bilirubin, and moreover a histopathological study was carried out in liver, pulmonary, and renal tissue fragments to identify any morphological changes.

## 2. Materials and Methods

### 2.1. Determination of Acute Toxicity

Ferulic acid derivatives were evaluated in vivo, determining the lethal dose 50 (LD50) on Swiss white mice [9–11]. The study was carried out in accordance with the current guidelines on the ethics and ethics of the study on laboratory animals (Law no. 206/27 May 2004, Eu/2010/63-CE86/609/EEC) and with the opinion of the commission of 07.01.2014 of the Research Ethics of "Grigore T. Popa" University of Medicine and Pharmacy, Iasi for acute oral toxicity tests.

Toxicological screening was performed on male, Swiss, white mice weighing between 18 and 28 g. They were housed in clean polyethylene cages (one cage per batch) and acclimatized for 7 d before the start of the experiment, with access throughout this period to food and water ad libitum. The ambient laboratory conditions were relatively constant during the experiment, the temperature being 23 ± 2 °C, with a cycle of 12 h of light and 12 h of darkness and a relative humidity of 40–70%. Prior to the start of the experiment, the mice were deprived of food for 24 h, receiving only water ad libitum.

Acute toxicity was assessed by the lethal dose 50 assay which consists of administering compounds in doses that increase in geometric progression to calculate the dose at which 50% of the mice included in the experiment died. The test compounds were administered in various concentrations (500–6500 mg/kg body weight) orally as emulsions in Tween 80 (Sigma-Aldrich, Taufkirchen, Germany), and the survival rate was monitored over a period of 24 h, 48 h, 72 h, 7 d, and 14 d after administration.

### 2.2. Determination of In Vivo Anti-Inflammatory Potential

#### 2.2.1. Acute Inflammation Model Induced with Carrageenan in Rats

The study of the anti-inflammatory effect on a model of acute inflammation was performed on white, Wistar breed, adult, male rats weighing between 140 and 220 g, which were raised under identical laboratory conditions and acclimated for 7 d before the start of the experiment under conditions of a relatively constant environment: temperature 23 ± 2 °C, humidity 40–60%, cycle of 12 h light/dark. Then, 18 h before the start of the experiment, the animals were deprived of food, receiving only water ad libitum. All the chemicals, Tween 80, ferulic acid, diclofenac sodium, indomethacin, carrageenan were purchased from Sigma-Aldrich Company, Taufkirchen, Germany. Acute inflammatory edema was induced by carrageenan, according to the working protocols described in the literature with some

modifications [12,13]. After weighing, the rats were divided into groups of 6 rats by body weight. The test compounds, azetidine 2-one derivatives of ferulic acid (**6a–f**) were administered by gavage in 1/10 doses of LD50 by dispersion in Tween 80. The study also included a group considered positive control, treated with ferulic acid; two reference groups, treated with diclofenac sodium (5 mg/kg body) and indomethacin (1.5 mg/kg body); and a control group in which the animals received only the vehicle used for the administration of the substances to be studied, Tween 80 (0.5 mL/100 g body) [14,15]. According to the working protocol described in the literature, before the administration of the compounds, the volume of the left hind paw of the rat was determined using the LE 7500 digital plethysmometer (Panlab, Barcelona, Spain) (Figure 1). Edema was induced by the intra-implantation, in the left hind paw, of 0.2 mL of 1% k-carrageenan suspension in physiological serum (sodium chloride 0.9 mg/mL). Immediately after the carrageenan injection, the compounds to be studied were gavaged, after which the volume of the left hind paw was measured at 2, 4, 6, and 24 h after administration.

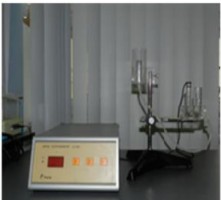

**Figure 1.** Plethysmometer LE 7500.

For the interpretation of the results, for each batch studied the difference between the volume of the hind paw of each rat at different time intervals (2, 4, 6, and 24 h) (Figure 2) and the volume initially determined, according to the calculation formula, was calculated:

$$\Delta V_{t/c} = V_{t/c} - V_i, \tag{1}$$

in which:

$\Delta V_{t/c}$ = volume of edema in the treated/control group;
$V_{t/c}$ = volume of left hind paw of rats in the treated/control group;
$V_i$ = volume of left hind paw of rats before the carrageenan administration.

The results were compared with those of the group treated with Tween 80, considered as the control group.

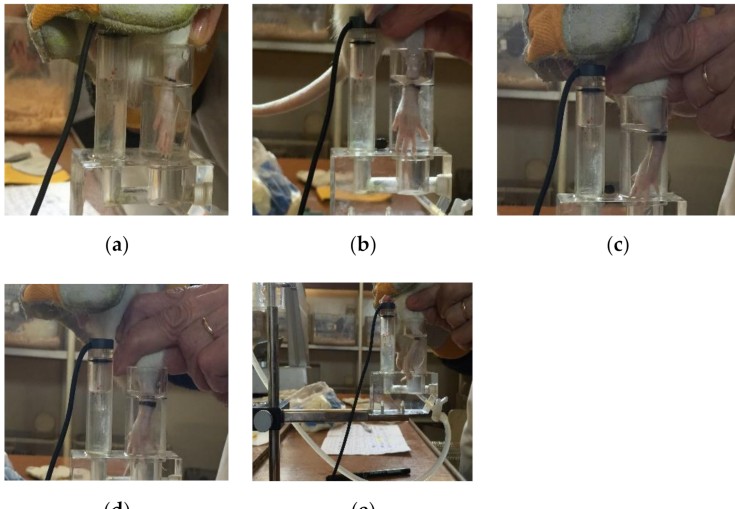

**Figure 2.** Images of rat paw measurement at different time intervals: (**a**) time $t_0$, before carrageenan injection; (**b**) 2 h after edema induction; (**c**) 4 h after edema induction; (**d**) 6 h after induction; (**e**) 24 h after the induction of edema.

To study the anti-inflammatory effect, for each time interval (2, 4, 6, 24 h) the percentage inhibition of acute carrageenan-induced edema was calculated according to the following calculation formula:

$$\% inhibition \ edem = \frac{(\Delta V_c - \Delta V_t)}{\Delta V_c}, \tag{2}$$

in which:

$\Delta V_c$ = volume of edema in the control group;
$\Delta V_t$ = volume of edema in the treated group.

### 2.2.2. Model of Chronic Inflammation Induced in Rats—Granuloma Test

The study of the anti-inflammatory effect of the azetidin-2-one derivatives of ferulic acid in a chronic inflammation model was performed using the granuloma test [16–19]. Work was performed on lots of 6 rats, Wistar breed white, weighing between 140 and 220 g, raised under identical laboratory conditions, and acclimated for 7 d before the start of the experiment under relatively constant environmental conditions: temperature 23 ± 2 °C, 40–60% humidity, 12 h light/dark cycle. Eighteen hours before the beginning of the experiment, the animals were deprived of food, with the water kept ad libitum. All the chemicals, ketamine, Tween 80, ferulic acid, diclofenac sodium, indomethacin, were purchased from Sigma-Aldrich Company, Taufkirchen, Germany. The rats were anesthetized with ketamine (80–100 mg/kg body) and then incised in the interscapulohumeral area; in the incision were introduced two sterile 60 mg cotton fiber pellets. The pellets were kept in the respective area for 7 d, during which time the experimental animals received by gavage, during the 24 h interval, the tested substances, azetidin-2-one derivatives, in a dose of 1/10 of LD50 by dispersion in Tween 80. The study also included a group considered a positive control, treated with ferulic acid; two reference groups, treated with diclofenac sodium (5 mg/kg body) and indomethacin (1.5 mg/kg body); and a control group in which the animals received only the vehicle used for the administration of the substances to be studied—namely, Tween 80 (0.5 mL/100 g body). After 7 d, the pellets were extracted, weighed in a wet state, dried for 18 h at 57 °C, and weighed again. The lower the mass of the pellet, the more intense the anti-inflammatory action of the studied compound is considered to be.

## 2.3. Determination of Liver Toxicity

The investigation of liver function was performed by determining the serum levels of liver enzymes: alanine aminotransferase (ALT), aspartate transaminase (AST), lactate dehydrogenase (LDH), and bilirubin (total and direct) [20–23]. The RX Imola 7201-0505 analyzer produced by Randox Laboratories LTD, London, United Kingdom was used for the biochemical determinations.

The enzyme activities were determined on serum, by spectrophotometric measurements after colour reactions or reactions based on UV detection. The kits are from Diamedix, Bucuresti, Romania and the reagents are in boxes which ensure their increased stability and the extended linearity of the results.

After 7 d of administration, rats were sacrificed to obtain blood samples. The animals were anesthetized with isoflurane, and blood samples of approximately 1 mL were collected from the abdominal aorta. The blood was placed in evacuated tubes containing K2-EDTA as an anticoagulant (BDTM Vacutainer, Becton Dickinson, Franklin Lakes, NJ, USA) and used for the plasma biochemical tests.

## 2.4. Histopathological Study

In order to evaluate the effect of the administration of the selected compounds in the form of suspension on the chronic inflammation, a histopathological study was carried out on liver, pulmonary, and renal tissue fragments to identify any morphological changes. At the end of the experiment, the animals were euthanized and fragments of liver, lung, and kidney tissues were taken from certain groups in the study [24–26]. The 1 cm liver, kidney, and lung samples were put in vials with 10% neutral formalin solution (Sigma Aldrich, Taufkirchen, Germany) and fixed at room temperature for 48 h. These were processed for inclusion in paraffin, followed by microtome sectioning and H&E staining. To examine the histological preparations thus obtained, a Nikon Eclipse 50i microscope was used.

## 3. Results

## 3.1. Determination of Acute Toxicity

The acute toxicity was assessed by the lethal dose 50 assay, the tested compounds being administered in various concentrations (500–6500 mg/kg body weight) orally as emulsions in Tween 80 and the survival rate was monitored over a period of 24 h, 48 h, 72 h, 7 d, and 14 d after administration. The Karber arithmetic method based on the following calculation formula was used to determine the LD50:

$$LD_{50} = LD_{100} - \frac{\sum (a+b)}{n},$$
(3)

in which,

a = the difference between two successive doses of administered substance;

b = average number of dead animals in two successive lots;

n = number of animals in a herd;

LD100 = lethal dose 100 (representing the amount of substance that causes the death of all animals in the experimental group).

The determination of lethal dose 50 (LD50) is a first step in assessing the toxicological profile of an active substance. Depending on the value of the lethal dose 50, the pharmaceutical substances could be classified into six classes: super toxic, extremely toxic, very toxic, moderately toxic, weak toxic, and practically non-toxic. The establishment of LD50 highlights the toxicological profile of the administered substance, being inversely proportional to its toxicity. The lower the LD50 value, the more toxic the test compound is. The LD50 values for the azetidine-2-one derivatives are shown in Table 2.

**Table 2.** Lethal dose 50 (LD50 in mg/kg body) for azetidine-2-one derivatives.

| Compound | R | LD50 (mg/kg Body) | Compound | R | LD50 (mg/kg Body) |
|---|---|---|---|---|---|
| **6a** | –H | 1187.5 | **1d** | –NO$_2$(2) | 1450 |
| **6b** | –F(4) | 1750 | **6e** | –Br(4) | 1650 |
| **6c** | –Cl(4) | 1690 | **1f** | –OH(2) | 1780 |
| **Ferulic acid** | | | | 2875 | |

From the analysis of the obtained results, presented in Table 2, it can be appreciated that all the studied derivatives can be classified in the category of substances with moderate toxicity, having values of LD50 in the range of 500–5000 mg/kg body.

### 3.2. Determination of In Vivo Anti-Inflammatory Potential

#### 3.2.1. Acute Inflammation Model Induced with Carrageenan in Rats

In experimental pharmacology, the test of acute inflammatory edema induced by carrageenan in rats is frequently used to evaluate the anti-inflammatory effect.

In this study, the azetidine 2-one derivatives (**6a–f**) were tested at a dose of 1/10 of LD50, the obtained results being analyzed in comparison to diclofenac sodium and indomethacin, used as reference substances. The used concentrations for all the tested substances are: **6a** (R = H) = 118, 75 mg/kg body weight; **6b** (R = 4-F) = 175 mg/kg body weight; **6c** (R = 4-Cl) = 169 mg/kg body weight; **6d** (R = 2-NO$_2$) = 145 mg/kg body weight; **6e** (R = 4-Br) = 165 mg/kg body weigh; **6f** (R = 2-OH) = 178 mg/kg body weight; ferulic acid = 287.5 mg/kg body weight; diclofenac = 5 mg/kg body weight; indomethacin = 1.5 mg/kg body weight.

Figure 3 graphically shows the obtained results for the volume of acute inflammatory edema induced in the rat paw at different time intervals (2, 4, 6, 24 h) for the groups treated with the studied compounds (**6a–f**); for the groups treated with ferulic acid, diclofenac sodium, and indomethacin; and for control (treated with Tween 80). It is appreciated that the lower the volume of acute inflammatory edema, the higher the anti-inflammatory effect is.

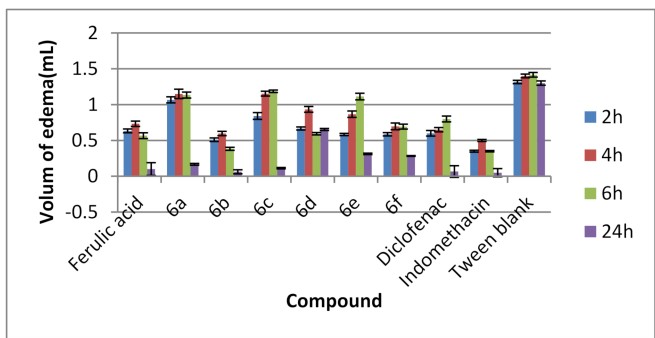

**Figure 3.** Variation in the inflammatory edema volume of the rat paw in the groups treated with 6 a–f derivatives (concentration being 1/10 of LD50), ferulic acid (concentration being 1/10 of LD50), diclofenac sodium (5 mg/kg body weight), indomethacin (1.5 mg/kg body weight), and the control group at 2, 4, 6, and 24 h. (Legend: concentrations for **6a** (R = H) = 118, 75 mg/kg body weight; **6b** (R = 4-F) = 175 mg/kg body weight; **6c** (R = 4-Cl) = 169 mg/kg body weight; **6d** (R = 2-NO$_2$) = 145 mg/kg body weight; **6e** (R = 4-Br) = 165 mg/kg body weight; **6f** (R = 2-OH) = 178 mg/kg body weight; Ferulic acid = 287.5 mg/kg body weight).

From the analysis of the obtained results it was found that, for all the studied compounds, the maximum anti-inflammatory effect—the reduction in the volume of the rat paw edema—was recorded 24 h after administration, similar to diclofenac and indomethacin, two established anti-inflammatory

drugs. Thus, it can be appreciated that the synthesized derivatives can be classified, from a pharmacokinetic point of view, as long-acting compounds.

The best results were obtained for the derivatives **6b** (R = 4-F) and **6c** (R =4-Cl), the effect of these compounds being comparable to diclofenac and indomethacin 24 h after administration. These statements are supported by the percentage values of acute inflammatory edema inhibition recorded for the studied compounds (**6a–f**) and calculated in relation to the value of the control group, which are presented in Table 3.

**Table 3.** Anti-inflammatory effect (% inhibition of inflammatory edema) of the tested compounds (**6a–f**) compared to ferulic acid, diclofenac sodium, and indomethacin at different time intervals.

| Lot/Compound | % Inhibition of Inflammatory Edema | | | |
| --- | --- | --- | --- | --- |
| | 2h | 4h | 6h | 24h |
| L1/**6a** | 18.98 ± 2.60 | 17.85 ± 5.14 | 20.03 ± 2.45 | 87.17 ± 6.33 |
| L2/**6b** | 61.23 ± 3.01 | 57.38 ± 2.80 | 72.94 ± 3.91 | 96.66 ± 6.20 |
| L3/**6c** | 36.11 ± 2.01 | 17.62 ± 0.50 | 16.24 ± 0.23 | 91.28 ± 8.05 |
| L4/**6d** | 49.36 ± 1.58 | 33.33 ± 1.41 | 58.11 ± 1.63 | 49.74 ± 1.12 |
| L5/**6e** | 55.69 ± 1.45 | 38.09 ± 1.98 | 21.41 ± 0.88 | 75.89 ± 2.42 |
| L6/**6f** | 55.44 ± 3.03 | 50.23 ± 2.91 | 51.05 ± 2.36 | 78.20 ± 1.51 |
| L11/Ferulic acid | 51.89 ± 2.20 | 47.61 ± 2.41 | 76.51 ± 9.27 | 92.30 ± 8.25 |
| L12/Diclofenac sodium | 54.43 ± 3.48 | 53.57 ± 2.43 | 43.52 ± 2.25 | 94.87 ± 11.61 |
| L13/Indomethacin | 73.41 ± 2.70 | 64.28 ± 1.81 | 75.29 ± 2.15 | 96.15 ± 11.10 |

### 3.2.2. Model of Chronic Inflammation Induced in Rats—Granuloma Test

In the chronic inflammation model induced by cotton fiber pellets, the weight of the wet pellets correlates with the transuded material while the weight of the dry pellets correlates with the amount of granulomatous tissue. Azetidine-2-one derivatives (**6a–f**) were tested at a dose of 1/10 of LD50. The tested concentrations for all substances are the same used in the acute inflammation model. The obtained results were analyzed by comparison with the results obtained for diclofenac sodium and indomethacin, used as reference substances, under the same experimental conditions.

Regarding the effect of the tested compounds on the formation of the granulation tissue correlated with the weight of the dry pellets, it was observed that all the studied compounds reduced the formation of the granulation tissue compared to the control group (treated with Tween 80); for one compound, the effect is comparable to that of indomethacin (Table 4).

**Table 4.** Effect of the azetidine-2-one derivatives (**6a–f**) on the proliferative process (granulation tissue formation) of chronic inflammatory edema induced in rats.

| Compound | R | Dose Administered (mg/kg Body/Day) | Average Weight of Dry Pellets (mg) | % Inhibition |
| --- | --- | --- | --- | --- |
| **6a** | -H | 118.75 | 0.623 | 20.53 |
| **6b** | -F(4) | 175.00 | 0.188 | 76.02 |
| **6c** | -Cl(4) | 169.25 | 0.330 | 57.90 |
| **6d** | -NO$_2$(2) | 145.00 | 0.570 | 27.29 |
| **6e** | -Br(4) | 165.00 | 0.524 | 33.16 |
| **6f** | -OH(2) | 178.00 | 0.452 | 42.34 |
| Ferulic acid | | 287.50 | 0.173 | 77.94 |
| Diclofenac sodium | | 5.00 | 0.153 | 89.49 |
| Indomethacin | | 1.50 | 0.147 | 81.25 |
| Tween 80 | | 0.5 mL/100 g | 0.784 | - |

The most important inhibitory effect of granulation tissue formation and consequently the most important anti-inflammatory effect was recorded for compound 6b (R = 4-F). This compound inhibited granulation tissue formation in 76.02%, the effect being comparable to that of indomethacin (81.25%). Under similar experimental conditions, ferulic acid showed a significant anti-inflammatory effect,

inhibiting the process of granulation tissue formation by 77.94%, the effect being comparable to that of indomethacin.

### 3.3. Determination of Liver Toxicity

The effect of the studied compounds, azetidin-2-one derivatives of ferulic acid (**6a–f**), on the liver function was evaluated by testing the activity of hepatic enzymes, alaninaminotransferase (ALT), aspartataminotransferase (AST), and lactate dehydrogenase (LDH), and the total and direct bilirubin determination. The obtained results are presented in Tables 5–7.

**Table 5.** Values of liver function parameters in rat lots **6a**, **6b**, **6c**, and **6d**.

| Biochemical Parameter | Lot/Compound | | | |
|---|---|---|---|---|
| | Lot 1/6a | Lot 2/6b | Lot 3/6c | Lot 4/6d |
| AST (UI/L) | 212.5 ± 1.48 | 198.5 ± 0.90 | 224 ± 0.70 | 185.5 ± 5.72 |
| ALT (UI/L) | 70.5 ± 0.35 | 48.5 ± 0.43 | 64 ± 1.83 | 52.5 ± 1.20 |
| LDH (UI/L) | 1826.5 ± 2.92 | 2342 ± 4.81 | 2546 ± 4.07 | 1866.5 ± 9.76 |
| Total bilirubin (mg/dL) | 0.105 ± 0.01 | 0.13 ± 0.01 | 0.115 ± 0.007 | 0.105 ± 0.01 |
| Direct bilirubin (mg/dL) | 0.045 ± 0.01 | 0.05 ± 0.014 | 0.033 ±0.014 | 0.03 ± 0.01 |

**Table 6.** Values of the liver function parameters in rats from groups 6e, 6f, FA and Diclofenac sodium.

| Biochemical Parameter | Lot/Compound | | | |
|---|---|---|---|---|
| | Lot 5/6e | Lot 6/6f | Lot 7/FA | Lot 8/Diclofenac |
| AST (UI/L) | 545 ± 7.70 | 244 ± 0.28 | 198.5 ± 2.47 | 133.5 ± 2.61 |
| ALT (UI/L) | 130.5 ± 5.86 | 58 ± 0.56 | 65.5 ± 1.48 | 49 ± 0.56 |
| LDH (UI/L) | 1181 ± 7.72 | 2734.5 ± 0.34 | 2020 ± 5.68 | 1116 ± 6.15 |
| Total bilirubin (mg/dL) | 0.13 ± 0.09 | 0.11 ± 0.014 | 0.11 ± 0.01 | 0.115 ± 0.02 |
| Direct bilirubin (mg/dL) | 0.045 ± 0.01 | 0.045 ± 0.007 | 0.055 ± 0.02 | 0.035 ± 0.01 |

**Table 7.** Parameter values of liver function in rats from groups Indomethacin, C1 $_{inflammation}$, Tween 80 and C2 $_{healthy}$, Tween 80.

| Biochemical Parameter | Lot/Compound | | |
|---|---|---|---|
| | Lot 9/Indomethacin | Lot 10 C1$_{inflammation}$/Tween 80 | Lot 11 C2$_{helthy}$/ Tween 80 |
| AST (UI/L) | 124.5 ± 0.35 | 269.5 ± 1.62 | 100 ± 1.08 |
| ALT (UI/L) | 59.5 ± 1.90 | 51 ± 1.41 | 46.15 ± 1.33 |
| LDH (UI/L) | 908.5 ± 4.16 | 1364 ± 4.78 | 400 ± 0.83 |
| Total bilirubin (mg/dL) | 0.175 ± 0.05 | 0.115 ± 0.04 | 0.085 ± 0.04 |
| Direct bilirubin (mg/dL) | 0.045 ± 0.01 | 0.042 ± 0.01 | 0.03 ± 0.02 |

In the series of azetidin-2-one derivatives, the liver enzyme concentration values were close to those recorded for the reference anti-inflammatories (diclofenac sodium and indomethacin) and slightly higher compared to the values for the healthy control group. The lowest hepatic impairment was recorded for the group treated with compound 6d (R = 2-NO$_2$). The values of direct and total bilirubin are comparable to the values of the healthy control group (Tables 5–7).

### 3.4. Histopathological Study

In order to evaluate the impact of the orally administrated compounds on different organs, liver, lung, and kidney fragments were examined. The histopathological study of these fragments, taken from each batch, allowed us to evaluate the changes induced by the chronic administration of azetidin-2-one derivatives of ferulic acid in suspension.

The examination of the control group of rats, for which no substance was administered, showed a normal morphology of the investigated organs, liver, lung, and kidney, respectively (Figure 4). The three organs were chosen for their particular sensitivity when substances in the category of those tested in this study are used.

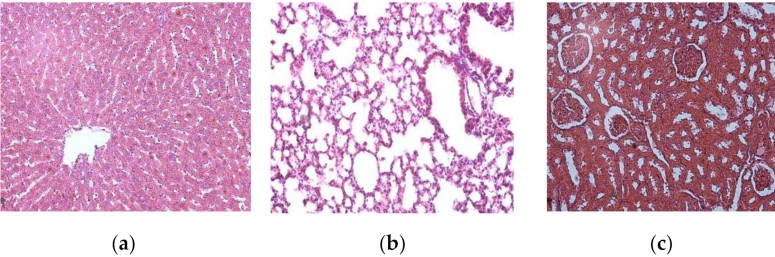

(a) (b) (c)

**Figure 4.** Morphology of the investigated organs for the control group: (**a**) liver normal morphology, col.HE, x20; (**b**) lung normal morphology, col.HE, x20; (**c**) kidney normal morphology, col.HE, x20.

The examination of the group treated with ferulic acid did not reveal any tissue changes (Figure 5); at the hepatic level, the hepatocytes maintained their characteristic structure and distribution, and the adjacent vascular system is intact. The pulmonary pattern is normal, without alterations, and at the renal level both the cortical and medullary show no structural changes.

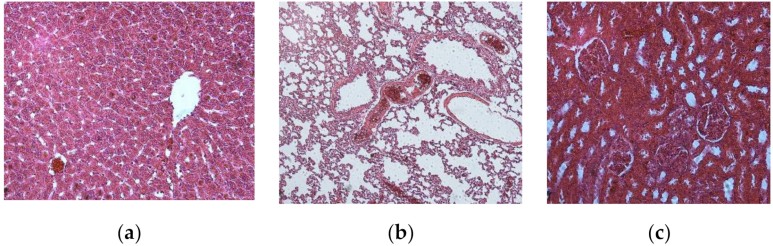

(a) (b) (c)

**Figure 5.** Morphology of the investigated organs for the ferulic acid-treated group: (**a**) liver normal morphology, col.HE, x20; (**b**) lung normal morphology, col.HE, x20; (**c**) kidney normal morphology, col.HE, x20.

The examination of the diclofenac-treated group revealed that, at the hepatic level, the hepatocytes maintain their structure but the sinusoids become dilated over large areas (Figure 6). At the pulmonary level, there is a massive destruction of the alveolar spaces, with hemorrhagic suffusions and alterations of the bronchial wallst as well as of the bronchiolar ones. The kidneys, in turn, show signs of tubular degeneration, marked by the presence of large areas of tubular necrosis.

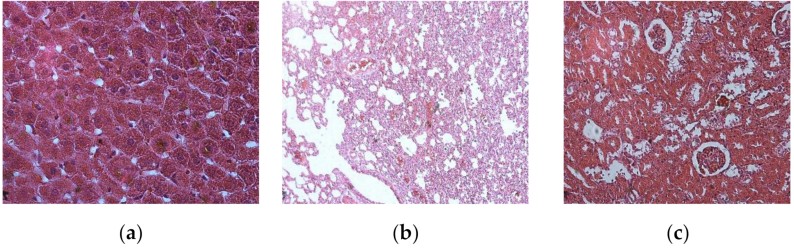

(a) (b) (c)

**Figure 6.** Morphology of the investigated organs for the diclofenac-treated group: (**a**) liver dilated sinusoids, col.HE, x20; (**b**) lung massive alveolar destruction, col.HE, x20; (**c**) kidney dilated tubular lumens, alterations of the tubular epithelium, col.HE, x20.

The examination of the group treated with Tween 80 revealed that there were no changes in the investigated organs cytoarchitectonics (Figure 7).

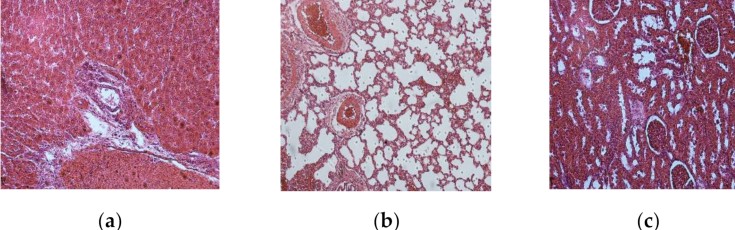

|       |       |       |
|:-----:|:-----:|:-----:|
| (**a**) | (**b**) | (**c**) |

**Figure 7.** Morphology of the investigated organs for the Tween 80-treated group: (**a**) liver normal morphology, col.HE, x20; (**b**) lung extensive vasodilatations, col.HE, x20; (**c**) kidney normal morphology, col.HE, x20.

The examination of the group treated with derivative **6b** (R = 4-F) showed the absence of alterations in the liver and renal tissue. Instead, the lung parenchyma showed a series of changes, represented by the distension of the bronchial walls, with hyperplasia of the respiratory epithelium, dilated vascular lumens, and inflammatory processes distributed around the main respiratory tract (Figure 8).

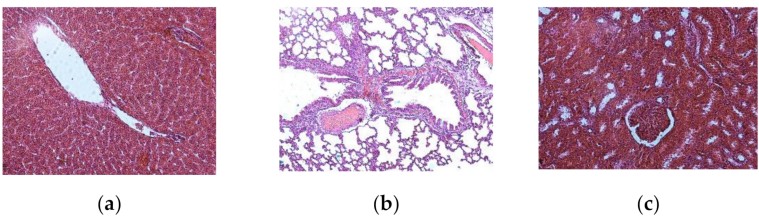

|       |       |       |
|:-----:|:-----:|:-----:|
| (**a**) | (**b**) | (**c**) |

**Figure 8.** Morphology of the investigated organs for the group treated with 6b (R = 4-F) derivative: (**a**) liver normal morphology, col.HE, x20; (**b**) lung bronchial distension, inflammatory reactions, col.HE, x20; (**c**) kidney normal morphology, col.HE, x20.

The absence of alterations in liver and renal tissue were found at the examination of all groups treated with azetidine-2-one derivatives. Additionally, an inflammatory processes in lung parenchyma was found in all the investigated groups treated with the studied derivatives.

## 4. Discussion

All chemical substances have toxicological risks, therefore safety tests are needed before any pharmacological validation can take place. Acute toxicity is mandatory according to standard guidelines and is necessary in the drug discovery and development process [10,11,27].

The basic compound of the series, compound **6a** (R = -H), showed the highest degree of toxicity. For this compound, the LD50 value was 1187.5 mg/kg body weight, the compound proving to be more toxic than ferulic acid (LD50 = 2875 mg/kg).

The substitution of the aromatic ring in the azetidin-2-one structure with various substituents has been associated with the reduced toxicity of the base compound of series (**6a**). Regarding the influence of the nature of the substituent on the degree of toxicity, it can be appreciated that the most favorable influence was exerted by the substituents 2-OH (compound **6f**, LD50 = 1780 mg/kg) and 4-F (compound **6b**, LD50 = 1750 mg/kg). These compounds have been shown to be about 1.5 times less toxic than **6a**. A favorable influence was also exerted by the 4-Cl substituent, the corresponding compound **6c** (LD50 = 1690 mg/kg) being about 1.4 times less toxic than **6a**. An equally favorable influence was exerted by the halogen –Br, the corresponding compound 6e (LD50 = 1650 mg/kg) being about 1.38 times less toxic than **6a**.

Based on the obtained results, it can be stated that the studied azetidin-2-one derivatives of ferulic acid belong to compounds with moderate toxicity.

At the same time, by analyzing the influence of structural modulations it was found that the reduction in toxicity is influenced by the nature of the radical substituting the aromatic nucleus present

on the azetidine-2-one structure, all of the resulting compounds being less toxic compared to the unsubstituted derivative (**6a**) and slightly toxic than ferulic acid.

The most favorable influence was obtained by substituting the aromatic ring with 2-OH in the orto position (compound **6f**) and with fluorine in the para position (compound **6b**). These compounds have been shown to be less toxic to the parent compound, unsubstituted on the aromatic ring (**6a**).

It is well known that in the inflammation process superoxide ion production is increased, pro-inflammatory cytokines as $\alpha$-tumor necrosis factor and interleukin-6 (TNF-$\alpha$ and IL-6) are produced, and also the cyclooxygenase-2 enzyme (COX-2) is involved [28–30].

In the last few years, researchers have found out that ferulic acid and its derivatives reduce xanthine oxidase and cyclooxygenase activity. They suggested that the anti-inflammatory mechanism might be achieved by inhibiting the COX-2 enzyme, similarly to NSAIDs [31]. Additionally, they claim that ferulic acid and its derivatives inhibit superoxide ion generation in macrophages and have shown good TNF-$\alpha$ and IL-6 inhibitory activity (up to 30–40% TNF-$\alpha$ and 60–75% IL-6 at 10 $\mu$M concentrations) [31–33].

Based on the results obtained on in vitro anti-inflammatory evaluation studies which were previously published [7,8] and from the literature data regarding the anti-inflammatory potential and mechanism of ferulic acid, our azetidin-2-one derivatives of ferulic acid were included in a pharmacological screening test aimed to determine the in vivo anti-inflammatory effect.

The biology of the test animals is generally similar to that of humans, and for that reason the animals served as models of human response [34].

Various scientific researchers have used in their in vivo studies mice as experimental animals for acute toxicity test, in order to establish the lethal dose 50 values and to highlight biological action other in vivo tests using rats. Thereby, the use of mice to calculate the lethal dose 50 has been reported, while the evaluation of the antidiabetic, antioxidant or anti-inflammatory potential was performed on Wistar rats, highlighting all the parameters and biological markers needed [35–38].

International guides (OECD 423/2001) states that both animal species, mice and rats, might be used in toxicity tests, both being rodents. Several scientists conducted experiments on both mice and rats in order to determine the lethal dose 50 and reported no significant differences, regardless of the experimental animal used [39,40].

Rats are usually used for studies of inflammation process which is evaluated by stimulants like carrageenan [41,42]. Edema and inflammation appears when carrageenan is injected. Different mechanisms of actions are involved leading to releasing of several mediators such as histamine, bradykinin, and serotonin by the cells during 1 h [43,44]. In the next 2–3 h, prostaglandins (PGE1, PGE2) responsible for acute inflammation are released [44,45].

Analyzing the obtained results, it was observed that 2 h after administration, the percentage of inhibition of inflammatory edema was between 18.98–61.23%, the most intense effect being recorded for compounds **6b** (R = 4-F, 61.23%), **6e** (R = 4-Br, 55.69%) and **6f** (R = 2-OH, 55.64%) their effect being slightly more intense than that of diclofenac (54.43%).

At 4 h after administration, the percentage inhibition of acute inflammatory edema was between 17.62% and 57.38%, the most intense effect being recorded for compound **6b** (R = 4-F, 57.38%), the effect being slightly more intense than that of diclofenac (53.57%). A comparable effect of diclofenac was also observed for compound **6f** (R = 2-OH, 50.23%).

After 6 h of administration, the percentage of inhibition of acute inflammatory edema was between 16.24–72.94%, compound **6b** (R = 4-F, 72.94%) showed an effect which was more intense than that of diclofenac (43.52%) and comparable with that of indomethacin (75.29%). Additionally, compounds **6d** (R = 2-NO$_2$, 58.11%) and **6f** (R = 2-OH, 51.05%) proved a very good inhibition percentage than that of diclofenac (43.52%).

To highlight the long-lasting effect, the anti-inflammatory effect was followed for 24 h. In this case, the percentage of inhibition of acute inflammatory edema was between 49.74% and 96.66%. Good results were obtained for derivatives **6b** (R = 4-F) and **6c** (R = 4-Cl), the effect of these compounds being

comparable to diclofenac and indomethacin 24 h after administration. The compound **6b** (R = 4-F) point out an inhibitory percentage of acute inflammatory edema of 96.66%, the effect being slightly more intense compared to diclofenac (94.87%) and comparable to indomethacin (96.15%). The compound **6c** (R = 4-Cl) presented an inhibitory percentage of acute inflammatory edema of 91.28%, comparable to diclofenac and indomethacin.

It should be noted that compound **6b** had an appreciable anti-inflammatory effect at 2, 4, 6, and 24 h after administration.

The analysis of the obtained results shows that for all the studied compounds, the maximum anti-inflammatory effect, respectively the reduction in the volume of rat paw edema, was registered 24 h after administration, similar to diclofenac and indomethacin. The most active compounds in the series was found to be **6b** (R = 4-F) and **6c** (4-Cl). At 24 h after administration, the effect of the compound **6b** (R = 4-F) was slightly more intense than that of diclofenac sodium, and for compound **6c** (4-Cl) the effect was comparable to the reference substances diclofenac sodium.

In the acute inflammation model, all the studied compounds showed a maximum anti-inflammatory effect 24 h after administration, which suggests that these compounds can be classified, from a pharmacokinetic point of view, in the category of long-acting compounds.

The cotton pellet granuloma method has been widely used to evaluate the transudative, exudative, and proliferative components of chronic inflammation, which is generated when the body fails to respond against inflammatory agents and fibroblast proliferation and formation of granulomatous tissues is installing [17,46,47]. Due to tissue injury, a cascade of cellular reactions is induced, and the release of pro-inflammatory cytokines followed by subsequent inflammatory reactions is initiated [18,48]. The transudative component is represented by plasma and blood figurative elements: leukocytes, monocytes, erythrocytes, and fibrin. These elements are found extravasated in a natural space (exudate) or in the interstitium (inflammatory infiltrate). At the same time, in the vast majority of inflammatory processes proliferative reactions are present, in varying degrees, accompanying or most often following alterative and exudative reactions. These are based on the proliferation of cells in the connective tissue; occasionally the proliferation of parenchymal cells can occur and, finally, the formation of granulation tissue takes place. In inflammation, granular tissue can form not only through a repair process and the organization of the exudate, but also as its own inflammatory phenomenon. This phenomenon occurs when there is a large proliferation of macrophages with mass formation in these cells that survive by nutrient intake through neoformation vessels [18,49].

Anti-inflammatory drugs such as indomethacin inhibit the release of inflammatory mediators, significantly prevent the formation of inflammatory edema, and suppress both exudate and granulation tissue formation as a result of subcutaneous cotton pellet implantation [18,50]. However, there are some side effects when there is prolonged use of the NSAIDs [46].

The significant inhibition of cotton pellet induced granuloma by our derivatives suggests an efficacy of them in inhibiting granulocyte infiltration and increase in the number of fibroblasts during granuloma tissue formation and, thus, a potential for their use in the treatment of chronic inflammatory conditions.

Regarding the effect of the tested compounds on the formation of granulation tissue, correlated with the weight of dry pellets, it was observed that all the studied compounds reduced the formation of granulation tissue compared to the control group (treated with Tween 80), with one of them exerting an effect comparable to that of indomethacin.

The most important effect of inhibiting the formation of granulation tissue and consequently the most important anti-inflammatory effect was recorded for compound **6b** (R = 4-F). This compound inhibited the formation of granulation tissue by 76.02%, the effect being comparable to that of indomethacin (81.25%). Under similar experimental conditions, ferulic acid showed an appreciable anti-inflammatory effect, inhibiting the formation of granulation tissue by 77.94%, the effect being comparable to that of indomethacin.

The case of the chronic inflammation model, it was observed that the studied compounds (**6a–f**)—including the compound with the most important anti-inflammatory action, **6b** (R = 4-F)—had a more intense effect in inhibiting the proliferative component, granulation tissue formation, than in the transudative component of chronic inflammation.

Considering that the liver is the main target of drug toxicity, hepatotoxicity is a frequent side effect, being the principal cause of new drug candidate failure [51,52]. Liver damage occurs since the biotransformation of drugs and external substances takes place in this very important organ of metabolism. Currently, over 1000 drugs are known to induce liver injury [53].

Liver enzymes activity can increase five to ten times in liver disease. Although ALT and AST are in high concentrations in hepatocytes, only ALT characterizes the normal functioning of the liver because AST is more present in the myocardium, skeletal muscle, brain, and kidneys [22]. Thus, ALT is the most commonly used indicator of hepatic cytolysis in the detection of even minor liver lesions. Lactate dehydrogenase is present in the form of five isoenzymes, of which only the LDH5 form has a higher specificity and may indicate elevated liver disease.

A well-functioning liver function leads to the metabolism and excretion of bilirubin, so serum bilirubin levels increase when bilirubin production exceeds its metabolism and excretion.

Taking into account its hepatoprotective effect, ferulic acid has been shown to decrease the elevated serum levels of the liver marker enzymes, AST and ALT, in rats subjected to ethanol-induced hepatotoxicity [3]. Additionally, azetidin-2-ones have shown good inhibitory activity on human chymase—a chymotrypsin-like serine protease which is thought to play an important role in chronic inflammation [54,55].

Due to the properties of ferulic acid and azetidin-2-ones, we expect that our azetidin-2-one derivatives of ferulic acid to have a good toxicological profile. The hepatotoxicity induced by the azetidin-2-one derivatives of ferulic acid was investigated based on blood parameters after oral administration for 7 d.

From the analysis of the obtained results, it can be seen that the liver enzymes (AST, ALT, LDH) recorded increased values for the group with chronic inflammation induced by the granuloma test (lot 10), compared to the healthy group (lot 11), which denotes the negative influence of inflammatory phenomena on liver function. The values recorded for lot 10 were 269.5 IU/L (AST), 51 IU/L (ALT), and 1364 IU/L (LDH) compared to the values of 100 IU/L (AST), 46.15 IU/L (ALT), and 400 IU/L (LDH) recorded for the healthy control group.

The lowest hepatic impairment was recorded for lot 4, treated with compound **6d** (R = 2-NO$_2$), and for lot 2, treated with compound **6b** (R = 4-F). In these cases, the liver enzyme concentration values were relatively close to those recorded for the reference anti-inflammatory drugs used in the study (diclofenac sodium and indomethacin) and slightly elevated compared to the values recorded for the healthy control group, but not enough to cause hepatotoxicity.

Regarding the concentration of ALT enzyme, considered the most relevant indicator of normal liver function (hepatic cytolysis indicator) [56], it was found that the least toxic for the liver were compounds **6b** (R = 4-F, lot 2) and **6d** (R = 2-NO$_2$, lot 4), for which the value of the ALT enzyme was 48.5 IU/L (**6b**) and 52.5 IU/L (**6d**), respectively, values comparable to those recorded for diclofenac sodium (49 IU/L), indomethacin (59.5 IU/L), and the healthy control batch (46.15 IU/L). In response to 1/10 of LD50, the ALT activity increased by 1.0–2.8 fold for the investigated compounds compared to that of the healthy control group.

For all the studied derivatives, the other parameters for detecting liver damage, the levels of AST and LDH, increased significantly by 1.8–5.4 fold and 2.9–6.8 fold, respectively. The LDH enzyme value was higher compared to the healthy control group (lot 11, 400 IU/L), the diclofenac-treated group (LDH = 1116 IU/L), and indomethacin (LDH = 908.5 IU/L).

The direct and total bilirubin values for the groups treated with the azetidin-2-one derivatives of ferulic acid (lot 1–6), the group treated with ferulic acid (lot 7), the group treated with diclofenac sodium (lot 8), and the group treated with indomethacin (lot 9) are comparable to the values of the

healthy control group (lot 11), for which the value of total bilirubin was 0.085 mg/dL and that of direct bilirubin was 0.03 mg/dL. The total bilirubin in the blood increased by 1.2–1.5 fold for the ferulic acid derivatives compared to that of the healthy control group.

However, all the studied compounds showed values of total and direct bilibubin comparable to the values recorded for the healthy control group, diclofenac sodium and indomethacin, which suggests a toxicological profile similar to that of classic anti-inflammatory drugs. We assume that the hepatotoxic mechanism of azetidin-2-one derivatives of ferulic acid is a mixed liver injury one.

The analysis of the toxic potential of a therapeutic agent on target organs is not complete without histopathological assessments. Liver, kidney, and lung microscopic pathology serve as important tools for identifying and characterizing organ injuries whether biochemical and macroscopic changes are identified or not.

Some of the main patterns of liver injury during hepatotoxicity include zonal necrosis and vascular lesions [57]. The general pathology of renal structures includes glomerular hypercellularity, the tubular necrosis being caused by manifestations of either local metabolic abnormalities or systemic processes [57]. The destruction of lung parenchymal tissue is morphologically characterized by an increase in the size and number of small fenestrae in alveolar walls, fibrovascular trabeculae breakdown, with the remodeling of acini leading to airspace enlargement [58]. Interstitial restrictive lung diseases are characterized by inflammation or the filling of the air spaces with exudate and debris [59].

In this study, azetidin-2-one derivatives of ferulic acid did not produce any detectable and meaningful change in the liver and kidneys, with some histopathological changes in the lung in rats at a 1/10 of LD50 dose just being noticed.

The histopathological changes in the lung showed an inflammatory process distributed around the main respiratory tract, hyperplasia of the epithelium, and dilated vascular lumens.

The histopathological study of Wistar rats of all six azetidin-2-one derivatives of ferulic acid caused some histopathological alterations in the lung such as inflammation, dilated vascular lumens, and bronchial distension at 1/10 of LD50. All these alterations are not so harmful compared to those observed for the diclofenac-treated group, where we noticed a massive destruction of the alveolar spaces with hemorrhagic suffusions and alterations of the bronchial walls. No alterations were found in the liver and kidney parenchyma.

It can be stated that the studied derivatives showed a lower toxicity than that of diclofenac sodium, which suggests a lower toxicological profile compared to this classic anti-inflammatory drug.

## 5. Conclusions

The in vivo biological potential of six novel azetidine-2-one derivatives of ferulic acid was evaluated in the present study. The acute toxicity study revealed that the studied derivatives belong to compounds with moderate toxicity. The anti-inflammatory potential was determined in a model of acute and chronic inflammation, the obtained results suggesting that these compounds may be classified in the category of long-acting compounds with an intense effect of inhibiting the proliferative component. The most active compound in the series was found to be compound **6b** (R = 4-F). A liver function evaluation showed that the AST, ALT, LDH, and bilirubin (total and direct) concentration values were close to those recorded for the reference drugs. Additionally, no major changes were found in the histopathological investigation. All the obtained data designate a lower toxicological profile than diclofenac sodium.

We can record that all six new azetidine-2-one derivatives of ferulic acid, and mainly compound **6b**, have promising anti-inflammatory activity.

**Author Contributions:** Conceptualization, C.D.S., C.E.L., C.L.Z., and L.P.; methodology, M.D., A.T.I, O.M.D., M.B., and L.P.; software, M.D. and C.D.S.; validation, C.E.L., C.L.Z., and L.P.; writing—original draft preparation, M.D. and C.D.S.; writing—review and editing, C.D.S.; visualization, C.E.L. and C.L.Z.; supervision, L.P. All authors have read and agreed to the published version of the manuscript.

**Funding:** This research was funded by "Grigore T. Popa" University of Medicine and Pharmacy Iasi, through internal research grant number 29244/20.12.2013.

**Conflicts of Interest:** The authors declare no conflict of interest.

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
