# Peer review of "Biological Evaluation of Azetidine-2-One Derivatives of Ferulic Acid as Promising Anti-Inflammatory Agents"

_processes, doi:10.3390/pr8111401_

Round 1

Reviewer 1 Report

Some comments:

The authors used Swiss whit mice for toxicity test, while using rat for inflammation test. Although mice and rat look similar but the toxicity cannot be fully translated. Please justify.

The authors has include proper positive controls in their studies (Indomethacin and diclofenac sodium), it is good that the authors can label clearly in their figures and legends that the concentrations of all tested compounds they used in each test.

The authors claimed that the derivatives of ferulic acid are potent anti-inflammatory agents, but I don't think it is potent when compared with Indomethacin they used as positive control. Please consider to refine some better description.

For figure 4b, the power of resolution seems different from other similar figures, please check.

The authors has published some in vitro results, I suggest they should mention in the introduction, based on their in vitro findings and extend to current animal study.

In the discussion, the authors should discuss some possible mechanism of action of the anti-inflammation activities of ferulic acid. Is it related NASID?

Reviewer 2 Report

Please consider the following recommendations:

  • the general structure for the compounds 6 (line 82) must be redrawn with a larger fonts for the atoms
  • nocardicins and monobactams (line 75), are the same subclass of beta-lactam antibiotics. There are no two different classes. 
  • there is only one sulbactam and only one tazobactam. No sulbactams and tazobactams (line 75)

Round 2

Reviewer 1 Report

No further comments.